# Replication Protein A, the Main Eukaryotic Single-Stranded DNA Binding Protein, a Focal Point in Cellular DNA Metabolism

**DOI:** 10.3390/ijms25010588

**Published:** 2024-01-02

**Authors:** Heinz Peter Nasheuer, Anna Marie Meaney, Timothy Hulshoff, Ines Thiele, Nichodemus O. Onwubiko

**Affiliations:** 1Centre for Chromosome Biology, School of Biological and Chemical Sciences, Biochemistry, University of Galway, H91 TK33 Galway, Ireland; 2Molecular Systems Physiology Group, School of Biological and Chemical Sciences, University of Galway, H91 TK33 Galway, Ireland

**Keywords:** DNA replication, DNA repair, homologous recombination, DNA damage signalling, replication protein A, DNA binding, protein interactions

## Abstract

Replication protein A (RPA) is a heterotrimeric protein complex and the main single-stranded DNA (ssDNA)-binding protein in eukaryotes. RPA has key functions in most of the DNA-associated metabolic pathways and DNA damage signalling. Its high affinity for ssDNA helps to stabilise ssDNA structures and protect the DNA sequence from nuclease attacks. RPA consists of multiple DNA-binding domains which are oligonucleotide/oligosaccharide-binding (OB)-folds that are responsible for DNA binding and interactions with proteins. These RPA–ssDNA and RPA–protein interactions are crucial for DNA replication, DNA repair, DNA damage signalling, and the conservation of the genetic information of cells. Proteins such as ATR use RPA to locate to regions of DNA damage for DNA damage signalling. The recruitment of nucleases and DNA exchange factors to sites of double-strand breaks are also an important RPA function to ensure effective DNA recombination to correct these DNA lesions. Due to its high affinity to ssDNA, RPA’s removal from ssDNA is of central importance to allow these metabolic pathways to proceed, and processes to exchange RPA against downstream factors are established in all eukaryotes. These faceted and multi-layered functions of RPA as well as its role in a variety of human diseases will be discussed.

## 1. Introduction

Replication protein A (RPA) is an ubiquitously expressed protein present in all eukaryotes [1]. It plays essential roles in a wide range of DNA metabolic activities, including DNA replication, DNA recombination, DNA repair, and DNA damage signalling [1,2]. RPA was initially identified in the Simian Virus 40 DNA replication system where it supported T-antigen-dependent DNA unwinding [3,4,5]. RPA efficiently and with high affinity binds to single-stranded DNA (ssDNA) to avoid formations of secondary structures such as hairpins and to protect ssDNA against nuclease attacks [1,6,7,8,9]. Thus, as the main eukaryotic ssDNA-binding protein, RPA safeguards ssDNA but also recruits various factors to the RPA–ssDNA complex such as ATRIP for ATR-dependent DNA damage signalling [2,10]. Protein recruitment allows replication, checkpoint signalling, and repair pathways to occur [10]. Post-translational modifications may lead to conformational changes in RPA, giving rise to additional functions and characteristics [2,11].

## 2. RPA Structure, ssDNA and Protein Interactions

RPA is a heterotrimeric protein composed of subunits RPA70, RPA32, and RPA14 with apparent molecular weights of 70 kDa, 32 kDa, 14 kDa, respectively, as determined by SDS gel electrophoresis (Figure 1 [1,5,12]). Each subunit contains at least one of the universally conserved oligonucleotide/oligosaccharide-binding (OB)-fold characterised by twisted β sheets and ⍺-helix capped ends. These OB-fold domains allow RPA to bind to ssDNA or target proteins involved in DNA metabolism and DNA damage signalling [10,13,14]. The three subunits bind together through a heterotrimerisation core located in DNA-binding domain-C (DBD-C) of RPA70, DBD-D of RPA32, and RPA14 (DBD-E) facilitated by hydrophobic interactions of ⍺-helices in their OB-folds in a synergistic manner, as represented in Figure 1A [12,13] and shown in the RPA structures (Figure 1B–E). RPA70 and RPA32 are the main DNA-binding sites, but there is some evidence that RPA14 also possesses DNA-binding activity [15]. RPA14 appears to coordinate RPA70 and RPA32 association, essential for RPA function, particularly in the case of shorter DNA lengths [15].

The DNA-binding domains (DBDs) A and B bind ssDNA with a groove composed by the loops L12 and L45 flanking the β strands β2 and β3 of the OB-fold (Figure 1B; [13]). These two binding sites each contain two highly conserved aromatic residues, that stack with DNA bases, and two conserved hydrophobic aa. When the linker regions between the DBDs are not occupied by DNA, they become susceptible to proteolysis [13,16]. DBD-C contains crucial zinc-binding motifs for RPA structural stability, and its ssDNA-binding capability (Figure 1B [12,13]). RPA binds to ssDNA in three modes, an 8–10 nucleotide, 15–23 nucleotide, and 28–30 nucleotide mode [13,14]. These modes are a mark for the flexibility of the RPA–ssDNA interactions and may allow the modulation of the RPA binding to ssDNA by external proteins such as RAD52 as previously discussed ([2,14,17,18]; for further discussion, see Section 6.2). Here, two to three nucleotides per DBD are enough for RPA to establish a tight association with ssDNA [7]. Binding starts with the 8–10 nt-binding mode, where DBD-A and DBD-B bind in a condensed form of RPA, which is followed by sequential elongation to a 12–23 nt-binding mode, which includes DBD-A, -B plus -C, and then RPA establishes a 28–30 nt-binding mode, where ssDNA sequences interact with DBD-A, -B, -C and -D [13,14]. Recent findings suggest a dynamic binding mechanism for RPA when physically interacting with ssDNA where the modulation of DBD-C may cause the handoff of DBD-A and DBD-B and the affinity of RPA to ssDNA is reduced, but also additional nucleotide sequences can be bound by RPA and its mobility on ssDNA increases [7,14,17].

It was previously shown that RPA preferentially binds to polypyrimidine sequences [1,6]. The dissociation constant for poly(dT) was measured as K_D_ = ~1 nM, whereas sequences with a mixed base content and a length of 34 or 57 are bound with lower affinity, K_D_ = 40 nM and 10 nM, respectively [1,6,8,9]. Recent studies using substrates with different oligo(dT) lengths and EMSA confirmed the modular binding approach suggesting that RPA binds dT_10_ and dT_14_ with micromolar affinities. RPA has an affinity of K_D_ = ~10 nM for dT_15_ and dT_20_, whereas substrates with 30 or more dTs are bound with affinities K_D_ = ~5 nM [19]. Moreover, RPA has high affinities to natural occurring telomeric oligonucleotide sequences with a length of 18 nucleotides or longer K_D_ = ~1 nM whether they contain a G-quadruplex (G4)-forming sequence or not. In contrast, the CST (CTC1-STN1-TEN1) complex, an RPA-like protein complex with ssDNA-binding and telomere synthesis function ([20,21] see also Section 5), binds to linear telomeric ssDNA with similar affinities as RPA (1 to 5 nM depending on length with the longer ssDNA, ≥40 nucleotides, being preferred [7]). However, in contrast to RPA, the CST–ssDNA interactions are inhibited by G4 formation, suggesting important roles for RPA telomeric DNA metabolism [7].

**Figure 1 ijms-25-00588-f001:**
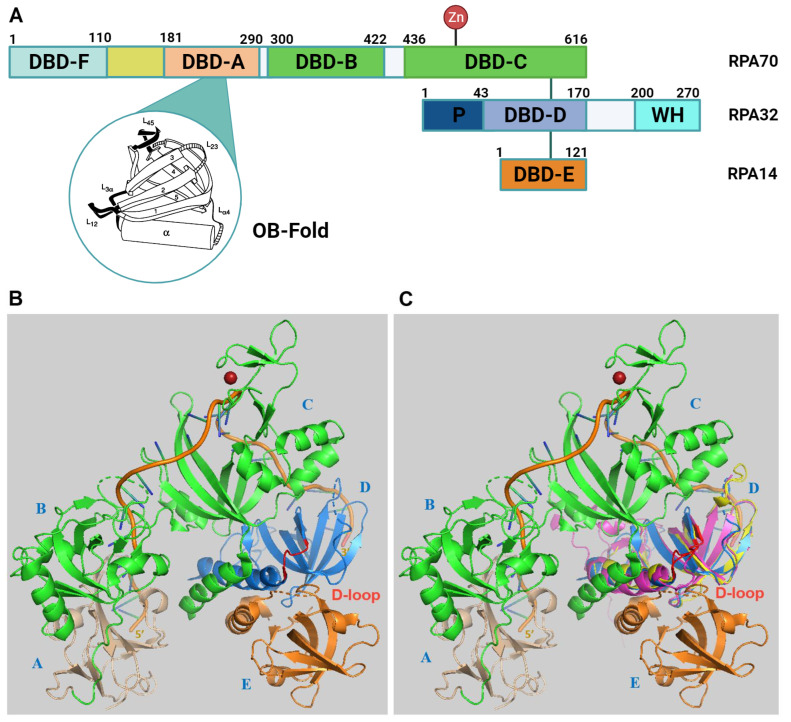
Structure of RPA. (**A**) The bars represent the three RPA subunits with DNA-binding domains (DBDs) DBD-A to DBD-F highlighted and their borders indicated by amino acid numbering of the human protein sequences [14,22]. Each DBD (colour code: RPA70: F (light brown), A (wheat), B and C (green); RPA32: D (light blue); RPA14: E (light orange)) contains a highly conserved OB-fold. The structure of one of the DBDs is highlighted in the bubble within the panel. The DBD domains A to D are able to bind to ssDNA. In contrast, the unstructured linker between DBD-F and A is shown in yellow, and the winged helix domain (WH) of RPA32 in light cyan are mainly protein–protein interaction sites. DBD-C contains the zinc-binding motifs represented by the protruding Zn. The N-terminus of RPA32, shown in dark blue and with the letter P, has recognition sites for multiple protein kinases including CDKs and PIKKs [23]. The lines between the three subunits represent the points of interaction between the subunits through the heterotrimerisation core (adapted from [12,24]). In the panel, the human RPA aa sequence is used for the numbering. (**B**) The experimental RPA structure presents an image of the RPA heterotrimer including RPA70 (with DBD-A in wheat plus DBD-B and C in green, the associated Zn atom is shown in red), RPA32 in marine blue, RPA14 in light orange, and oligo(dT) ssDNA in orange with the 5′- and 3′-end marked (structure derived from PDB 4gop is in the 30 nt-binding mode [13] and presented using Pymol (Schrödinger Inc. (USA)). In panel (**C**), human RPA32 and STN1 OB-fold structures were predicted by AlphaFold [25] and are shown in yellow light and magenta, respectively. STN1 is a subunit of the RPA-like CST complex and is involved in the initiation of G strand synthesis at telomeres ([20,21], see also Section 5). Both human structures were aligned to the experimental *Ustilago maydis* RPA (structure derived from 4gop) using Pymol (for a better overview, aa N93 to T124 of STN1, which do not align to any structure, were omitted). The loops containing D157 of STN1, D151 of human RPA32, and D155 of *Ustilago maydis* RPA32 are shown in red. Panels (**D**,**E**) present images of the RPA trimer predicted by AlphaFold [25] without DNA and in a similar position as *Ustilago maydis* RPA in panels (**B**,**C**). DBD-F, which is lacking in the *Ustilago maydis* RPA structure, and DBD-A are in light green and light brown, respectively. The linker between the two domains is coloured in yellow. DBD-B, DBD-C and the short linker between the two domains are in green. The WH domain of RPA32, which is not shown in the *Ustilago maydis* RPA structure, is presented in cyan. DBD-D plus the linker between DBD-D and WH are shown in light blue. The loop in DBD-D containing the conserved D151 is called the D loop (see also panels (**B**,**C**)) and is shown in red. It is important to note that the linker regions of RPA70 and RPA32 plus the N-terminal phosphorylation sequences of RPA32 have a very low per-residue model confidence score (pLDDT) and might be unstructured. To allow a better presentation of the RPA complex, the N-terminus of RPA32, aa 1–33, was omitted in panels (**D**,**E**) but is shown in the Appendix A. In panel (**E**), the OB-fold of STN1 shown in light pink is aligned to human RPA using PyMol (for a better overview, aa N93 to T124 of STN1, which did not align to any structure, were omitted in the presentation). The loop containing the conserved D157 (D loop) is presented in red. The RPA domains are marked with blue letters, A to F (DBD-A to F) and WH in the structures.

To fulfil its functions, RPA not only binds to ssDNA but also to more than 50 proteins involved in DNA replication, repair, and recombination plus DNA damage signalling [10,14,26]. Here, studies of the protein–protein binding sites of RPA have revealed that DBD-F and adjacent N-terminal sequences, which include the unstructured linker region and DBD-A, form a hub for protein interactions (Figure 1A, and for their structural presentation, see Figure 1D, [10,14,22,26,27]). The N-terminal domain of RPA70, DBD-F, is involved in the binding of a variety of proteins in DNA replication, DNA damage response (DDR), and checkpoint signalling such as helicase B, WRN (Werner helicase), BLM (Bloom helicase), Tp53 (tumour-suppressor protein p53), RAD9 of RAD9–HUS1–RAD1 (9-1-1), ATR (ATM-Rad 3-related protein) ATRIP (ATR-interacting protein) and ETAA1, which is a newly identified ATR activator [10,14,26,28,29,30]. Many of these contacts are mediated by the side pocket and the basic and hydrophobic groove of DBD-F [10]. The adjacent linker region and DBD-A of RPA70 (see Figure 1A,D) bind additional proteins such as RAD51, RAD52, Polα-p180, SV40 Tag (SV40 T antigen), and XPA [14,26,31,32]. The WH domain of RPA32 is a second protein interaction hub of the RPA complex (see Figure 1A,D) and binds to various proteins such as the DNA polymerase α (Pol α) subunit Prim2, RAD52, Tag, UNG protein, and XPA [26,33,34,35]. Interestingly, the two protein binding hubs seem to be located on the same site of the RPA complex as predicted by AlphaFold (Appendix A). Additionally, the binding of RPA to human CDC45 and Primase-polymerase (hPrimpol1) regulates their cellular functions [9,36,37,38].

## 3. Alternative RPA and Its Emerging Functions in Neurodegenerative Diseases

In addition to the subunits of ‘canonical’ heterotrimeric RPA, researchers have found in primates a RPA32-related protein, RPA4, a 30 kDa protein, which has 47% aa sequence identity and a 63% aa similarity with human RPA32 and can replace the latter in a complex with RPA70 and RPA14, forming a protein complex called alternative RPA (Alt-RPA) [39,40,41]. RPA4 is mainly expressed in non-dividing cells and is preferentially detected in the bladder, esophagus, lung, placental, prostate, and colon mucosa tissue, but it is also found in human brain tissue [39,40]. Interestingly, only primates express the RPA4 protein but not mice [41,42]. Most biochemical activities of Alt-RPA are similar to those of canonical RPA. Alt-RPA stimulates SV40 Tag during the origin-dependent unwinding of dsDNA and DNA polymerase δ (Pol δ) as well as canonical RPA [43,44]. However, Alt-RPA has weaker interactions with Pol α and is unable to stimulate DNA synthesis and de novo priming by Pol α during SV40 DNA replication in vitro [43,44]. Additionally, Alt-RPA does not support cellular DNA replication [40,41,44]. In contrast, Alt-RPA stimulates DNA synthesis by Pol α in the presence of RFC/PCNA, which is again contrary to canonical RPA. Interestingly, Alt-RPA supports nucleotide excision repair and homologous recombination [40]. These findings suggested that since Alt-RPA has a widely overlapping protein interaction pattern with canonical RPA, its expression in cells might be relevant for cellular DNA repair functions or yielding a disease-relevant phenotype [40,42,44]. Recent findings backed the later hypothesis, and Alt-RPA expression supports CAG expansions in neurodegenerative diseases whereas canonical RPA suppresses them, suggesting a yin–yang relationship between Alt-RPA and canonical RPA in the modulation of CAG repeat instability [42]. The opposite roles of canonical and alternative RPA in neurodegenerative disease development including Huntington’s disease development may also explain the conundrum that humans with CAG repetition numbers of 36 or more in the Huntington’s disease gene, the Huntingtin gene, show a Huntington’s disease phenotype at some stage in their life. In contrast, mice with 50 CAG repeats have a Huntington’s disease-free life and even have several advantages in comparison to wild-type mice, including higher life-span expectancy and cognitive advantages [45,46,47,48]. Importantly, mice require 81^+^ CAG repeats in the Huntingtin gene to show a Huntington’s disease phenotype [47,48]. Since human express functional RPA4 and produce Alt-RPA but mice do not [39,40,42], the presence and absence of Alt-RPA may play a role in the different requirements of CAG repeat numbers for Huntington’s disease onset in these two organisms, possibly by modulating somatic CAG repeat expansions that modulate the disease state (B. Lahue, personal communication).

## 4. Roles of RPA in DNA Replication

DNA replication is a process that requires the precise duplication of an organism’s genetic material once and only once in each cell division cycle [49,50]. This process ensures that the daughter cells receive identical copies of the genetic material with each cell division, thereby maintaining genomic stability throughout numerous cell divisions [50]. RPA plays a central and essential role in eukaryotic DNA replication through its ssDNA binding and protection plus protein recruitment functions [49]. During the replication process, the MCM2-7 (minichromosome maintenance 2 to 7) complex is loaded onto origins of DNA replication with the help of ORC1-6 (origin recognition complex 1 to 6), Cdc6 (cell division cycle 6), and Cdt1. The activation of the MCM2-7 requires the loading of Cdc45 plus the GINS complex to chromatin and the formation of the CMG complex, the eukaryotic replicative helicase, which is supported by the proteins DONSON and DNA polymerase ε (Pol ε) [51,52,53,54,55,56,57,58]. At the beginning of the S phase, CMG helicase unwinds dsDNA at the origins, and the dsDNA is separated into two ssDNA templates to allow the DNA polymerase to have access to the genetic information and to allow the duplication of DNA (summarised in Figure 2 [49]). With the help of Cdc45, RPA binds to the exposed ssDNA in a polar fashion and stabilises the ssDNA [9,37,49,59]. Due to the polar orientation of the DNA, at replication forks, the unwound DNA strands give rise to template DNAs with a 5′–3′ and a 3′–5′ direction. Together with the knowledge that DNA polymerases only synthesize DNA in 5′–3′ direction with the template having a 3′–5′ directionality, the DNA is synthesised continuously, forming the leading strand, whereas the second strand is synthesised discontinuously in small pieces called Okazaki fragments and forms the lagging strand as presented in Figure 2 [49].

No replicative eukaryotic DNA polymerase starts DNA synthesis de novo. Therefore, a special enzyme, primase (often also called DNA primase), an RNA polymerase, synthesises short oligoribonucleotides with a length of ~10 nts or multimers thereof on ssDNA templates including unwound origin DNA to start DNA synthesis. Primase is an essential enzyme for DNA replication and functions as a dimer consisting of PRI1/Prim1/p49, the catalytic subunit, and PRI2/Prim2/p58, the regulatory subunit, which controls primer lengths [62,63,64]. The primase dimer forms a heterotetrameric complex with the two larger subunits PolA1/p180 carrying the catalytic DNA polymerase activity and PolA2/p68/p70/B subunit, the second regulatory subunit, which is called DNA polymerase ⍺ (Pol ⍺) or DNA polymerase ⍺-primase [65]. Pol α is the only replicative DNA polymerase complex having primase activity and thus is central for DNA replication processes. Under special circumstances, such as replication fork restart, Primpol with the help of RPA may take over the restart function [36,66]. However, Pol α in association with the CST complex may also function in the replication restart [67]. Primer synthesis is initiated by the cooperation of CMG, AND-1-Ctf4-WDHD1, RPA, Pol ⍺ and primase at 5′ ssDNA (Figure 2 [52,53,68]). After the initiation, PRI2/Prim2 hands over the primer to the PolA1/p180 subunit of Pol ⍺ to elongate the primer by a DNA chain of 20 nucleotides. RPA serves as a ‘fidelity clamp’ of Pol α to stay on the ssDNA, suggesting that RPA and the largest subunit collaborate in the polymerase transition from primase to the large subunit of Pol ⍺, PolA1/p180 [32,62,69]. After the DNA synthesis of ~20 nts, Pol ⍺ leaves the template and is replaced by replication factor C (RFC) in contact with RPA. The former then uses ATP hydrolysis to add proliferating cell nuclear antigen (PCNA) to the primed template [60,70]. In the next step, the 2nd polymerase transition, Pol δ is recruited and utilises PCNA as a loading clamp to elongate the newly synthesised DNA strand in a processive manner. For the leading strand synthesis, in a 3rd polymerase transition, the short RNA–DNA primer is then handed over to Pol ε, which is in close contact with the CMG helicase on the leading strand template and is supposed to support unwinding activity of CMG [51,71]. On the leading strand template, Pol ε synthesises the leading strand in a processive and continuous manner up to a size of a replicon [65,72]. Both Pol δ and ε are also heterotetrameric enzyme complexes, but they do not have primase activity associated [65,72]. In contrast to Pol α, which does not have 3′-5′ proofreading exonuclease activity, the Pol α exonuclease domain is inactive; both Pol δ and ε contain a DNA polymerase and 3′-5′ proofreading exonuclease activity in their largest subunits, Pol D1/p125 and PolE/Pol2/p260 [65,72]. Interestingly, single molecule experiments have shown that one Pol ε, one Pol δ and one to two Pol α bound to CTF4 are functional, collaborating at a replication fork [73,74].

During lagging strand synthesis, each Okazaki fragment is initiated by Pol α as described above. Taking the Okazaki fragment size and the human genome size into account, Pol α synthesises approximately 20 to 40 million primers to produce a complete round of lagging strand synthesis per cell division cycle, which is an enormous task [75]. In the following, the RNA primer is handed over from the primase to PolA1/p180 by PRI2/Prim2 with the support of RPA, whereas the RNA–DNA primer is then transferred from Pol α to Pol δ [65,70,72]. The latter synthesises the Okazaki fragment until it reaches the RNA moiety of the previous Okazaki fragment. Importantly, RPA needs to protect exposed ssDNA until pol δ encounters the next primer, and a DNA ligase ligates the fragments together [61,72]. After reaching the previous Okazaki fragment, Pol δ continues the synthesis of the associated Okazaki fragment into the previous fragment in a strand displacement mode. Thus, by removing the RNA primer and a part of the Pol α-synthesised DNA, the enzymes create a 5′-flap DNA, which for efficient ligation of the two Okazaki fragments needs to be cleaved off [61,72]. The enzyme flap endonuclease 1 (FEN1) takes over this role. FEN1 binds to the 5′-flap DNA and cleaves the DNA precisely at the ssDNA–dsDNA transition, leaving a nicked dsDNA remaining, which is an ideal substrate for LIG1 ligation [61,72]. Alternatively, RPA binds the 5′-flap ssDNA, protects the flap-DNA against FEN1 and recruits the endonuclease DNA2 to the ssDNA, which cleaves the flap overhang [61,76]. Finally, LIG1 ligates the nicked DNA, producing a new continuous dsDNA molecule. Interestingly, FEN1, LIG1, and Pol δ can associate with the same PCNA molecule during Okazaki fragment synthesis and maturation, occupying different regions of the PCNA homotrimer [61,72].

## 5. The Role of the RPA-Related, ssDNA-Binding Protein Complex CST in the Initiation of Okazaki Fragment Synthesis

To understand the initiation of Okazaki fragment synthesis and to elucidate the role of RPA in this process, it is important to know that primase has lower affinities for ssDNA than RPA, and it has been shown that RPA efficiently inhibits the primase activity of Pol α on natural ssDNA templates [77,78,79,80,81]. In the SV40 DNA replication, a mechanism for the reversion of the inhibition has been described. Similar to the interaction of RAD52 with the RPA32 WH domain, Tag interacts with the WH domain, reverses the RPA inhibition, and allows primase to synthesise primers on RPA-bound ssDNA [78,82,83]. One of the main functions in this process is the remodelling of the Pol α–primase complex from an inactive state to a priming active state [21,75,84,85,86]. This process is best understood in the initiation process of Okazaki fragment synthesis on the G strand of telomeres as part of the C strand synthesis [21,75,84,85,86]. The structure of Pol α in association with the CST complex, an RPA-like complex also called alpha-accessory factor (AAF) [87], shows that the two protein complexes have numerous interactions with each other and some might be important for the remodelling of the Pol α complex to a pre-initiation complex [21,75]. Additionally, these Pol α–CST–ssDNA structures show that in the complex, the CST binds the template ssDNA and directs this ssDNA toward the catalytic centre of PRI1/Prim1 [21,75]. Considering the structural similarity of RPA and CST, it is thought that RPA has similar functions as CST during the initiation of DNA synthesis on ssDNA [21].

Recently, it was shown that the STN1 subunit of CST and RPA32 alone are sufficient to stimulate Pol α initiation activity and increase its primase activity on ssDNA [77,88]. The OB-fold domain of human STN1 is sufficient to provide the stimulation of Pol α, and here, the conserved aa D157 of STN1, which is equivalent to D151 in human RPA32, is important for the stimulatory activity, whereas the ssDNA-binding activity of STN1 is not important [77]. In the published pre-initiation complex of CST and Pol α, D157 interacts with the C-terminus of PolA1 (S1365 and R1366) and might be important for the remodelling of Pol α from an inactive to an active form and thus provide the stimulatory function for Pol α during the primase reaction [75,77,84]. The crucial aa D157 of human STN1 is conserved in human and fungal RPA32 (D151 in human RPA32 and D155 in RPA32 of *Ustilago maydis* (Appendix A)) and is part of a loop between a β-strand and an α-helix structure (Figure 1B–E, Appendix A). Overlaying the OB-folds of RPA32 and STN1 with the published structure of the *Ustilago maydis* RPA complex shows that the D-containing loop (D-loop) is conserved between STN1 and these two RPA32 structures (Figure 1C [13]), which is also seen in the alignment of the STN1 OB-fold with the AlphaFold-predicted human RPA structure (Figure 1E and Appendix A). In RPA, this D-loop lies in a cleft between RPA70 and RPA14 (Figure 1B–E), which may prevent access to Pol α. Thus, the interaction of a mediator protein such as Tag with the RPA32 WH domain, which has been extensively studied by Fanning and co-workers [78], might open the cleft and allow the binding of DBD-D in RPA32 to the PolA1 C-terminus and support the remodelling and activation of Pol α [75]. Predictions of protein–protein interactions by the Walter group using AlphaFold-predicted protein structures also suggest an interaction of RPA32 D151 with the C-terminus of PolA1, which is the largest subunit of the Pol α complex [25,89]. These findings suggest that interactions of RPA32 and STN1 with PolA1 C-terminus as discussed here plus the previously suggested delivery of the ssDNA template to the catalytic domain of PRI1/Prim1 might be important for the initiation reaction by Pol α and the stimulation of primase activity [21,75].

## 6. Functions of RPA in DNA Damage Response Pathways

DNA replication is a complex process, and errors commonly occur despite the proof-reading exonuclease activity associated with the replicative polymerases Pol δ and ε [65]. Additionally, the genetic information of cells is under constant pressure from intracellular and extracellular/environmental stresses causing DNA damage. To preserve genomic stability, eukaryotic cells have developed multiple DNA repair pathways depending on the type of DNA damage to correct damaged DNA [65]. Moreover, conserved DNA damage signalling pathways are present in eukaryotic cells to maintain genome stability. All these processes require RPA to fully function [14,17,22,65,90].
Base excision repair (BER) involves the repair of damaged or modified bases in the DNA [65,90].Nucleotide excision repair (NER) is required for the removal of bulky DNA lesions induced, e.g., by UV radiation or chemicals [65,90].Mismatch repair (MMR) is responsible for the identification and elimination of mispaired nucleotides after DNA replication [65,90].Double-strand breaks (DSBs) repair has evolved in eukaryotic cells as two main pathways, homologous recombination (HR) and non-homologous end joining (NHEJ). HR utilizes a homologous DNA template to allow for the accurate repair of DSBs whereas NHEJ only re-joins the broken ends, opening the possibility of the induction of insertions or deletions [65,90].

### 6.1. RPA in DNA Damage Signalling

Since DNA lesions are detrimental to eukaryotic cells and the stability of their genetic information, a fast and precise response to damaged DNA is required to correct DNA lesions in the chromosomal DNA of cells to avoid the appearance of genetic diseases such as cancer [2,10]. Here, under environmental or cellular stress, the protein DOCK7 promotes the accumulation of RPA at chromatin and replication forks [91]. In addition to its ssDNA-binding capabilities, RPA uses its protein-binding abilities to recruit certain factors to damaged sites to directly function there or initiate downstream pathways against DNA damage [2,92]. Here, replication fork stalling causes RPA binding to ATRIP to recruit ATR to the RPA-bound ssDNA stretches [93]. ATR in turn phosphorylates downstream targets such as CHK1 and p53 to delay the cell cycle, permitting the stabilisation of the replication fork [2]. Additionally, at stalled replication forks, hPrimpol1 directly interacts with RPA70’s N-terminal domain, DBD-F, and this binding is essential for hPrimpol1 to restart cellular DNA synthesis at DNA lesions [10,36,66].

During DNA damage, the three RPA subunits are ubiquitinated. Specifically, RPA32 is modified by K63-linked ubiquitin chains which are important for ATRIP recruitment and ATR kinase activation [2]. DOCK7 is one protein that is phosphorylated by this ATR activation. In turn, Dock7 phosphorylation increases RPA association to chromatin, which again allows further ATR activation, essentially creating a positive feedback loop [91]. Cyclin-dependent kinases (CDKs) and phosphatidylinositol-3 kinase-related kinases (PIKKs) such as ATR, ATM (ataxia telangiectasia-mutated), and DNA-PK (DNA-dependent protein kinase) phosphorylate serine and threonine residues in the RPA32 N-terminal region during the cell cycle and in response to genotoxic stress [2,27,94,95]. There are eight phosphorylation sites in this RPA32 sequence where CDK phosphorylates S23 and S29, which then stimulates the ATR-dependent phosphorylation of S33. In turn, this causes the subsequent phosphorylation of the other threonine and serine residues by DNA-PK and ATM. Here, S4, S8, and T21 phosphorylation are hallmarks for replication fork breakage, and this hyperphosphorylation acts as a marker for resection at double-strand break sites [2,27,94,95]. Furthermore, during the DNA damage response, Wiskott–Aldrich syndrome protein (WASp) helps to coordinate the ssDNA binding of RPA and to manage DNA stress response more optimally [96]. The association of WASp at DNA damage sites results in stable RPA–ssDNA complexes that are important for efficient DNA repair [96].

### 6.2. RPA in DNA Repair Pathways

To avoid genome instability by bulky DNA adducts or major DNA structure distortions, nucleotide excision repair (NER) is an important DNA repair pathway (summarised in Figure 3 [97,98,99]). The importance of the NER pathway is underlined by the existence of a rare genetic disorder called Xeroderma pigmentosum (XP), in which genes coding for proteins involved in the NER pathway are mutated [98]. These proteins are called XPA to XPG and XPV, which is a DNA polymerase called Pol η [98,99]. In global genome NER, XPC and RAD23B first recognise DNA lesions; then, the ten subunit complex TFIIH is recruited, where XPB translocase and XPD helicase unwind the dsDNA [99]. XPA and RPA join the complex stabilising the ssDNA. These two proteins are also responsible for correctly positioning the ERCC1-XPF and XPG endonucleases. When ERCC1-XPF incises DNA at the 5′ end, RPA helps to recruit RFC for PCNA loading and DNA synthesis by Pol δ and/or ε (Figure 3). XPG allows 3′ incision to occur so a ligase can bind the DNA once the DNA synthesis is completed and the damaged DNA strand is replaced [99].

In contrast, small DNA lesions such as oxidative, deamination and alkylation lesions, which cause considerably less distortion in the DNA helix than NER (see above), are repaired by BER [97]. BER starts with enzymes, called DNA glycosylases, which cleave the bond between deoxyribose and the modified DNA base, removing the damaged base and leaving an abasic site (loss of nucleobase by hydrolysis) behind. Then, AP endonuclease, e.g., APE1 in humans, recognises these abasic sites and cleaves the 5′-phosphodiester bond on the abasic site, which leaves a deoxyribose at the 5′ end [97]. In following process, called short-patch BER, DNA polymerase β (Pol β) is the major enzyme involved in the process. Two domains of Pol β, deoxyribose phosphate lyase and the DNA polymerase domain, are involved in the repair process. The former catalyses the release of the 5′-deoxyribose unit, and the latter fills in the abasic site with the correct nucleobase. Finally, DNA ligase 1 or 3 ligates the nicked DNA [97]. As an alternative to the described short-patch repair, long-patch BER takes place. DNA polymerases Pol β or Pol δ/ε, depending on the proliferation state of the cell fill in the abasic site with the correct nucleobase, extend the DNA synthesis step beyond the abasic site and synthesise DNA in a strand-displacement mode. FEN1 then removes the produced 5′ flap, and LIG1 ligates the nicked DNA [97]. In BER, uracil-DNA glycosylase (UNG), an important BER protein, interacts with RPA via the WH domain of RPA32. RPA’s exact role in BER is unclear, but it is thought that RPA is involved in the gap-filling stage, similar to NER [2].

The MMR pathway repairs base–base mismatches and deletions or insertions that occur during the replication process and increases the fidelity of the replication process by a factor of at least 100 [97,100]. Mutations in MMR proteins lead to a hypermutation phenotype of organisms and may cause an early onset of cancer [97,100]. In 5′ nick-directed MMR, the complex consisting of MutSα or MutSβ heterodimers together with MutLα, RPA, Exo1 and HMGB1 (high mobility group box 1) binds to mismatched DNA and initiates the excision repair [97,101]. The EXO1 exonuclease then excises the damaged DNA, and RPA stabilises the resulting ssDNA gaps and becomes hyperphosphorylated [97,101]. Subsequently, at the 3′ end of the DNA at the dsDNA–ssDNA transition, RFC–PCNA complexes support the association of Pol δ, which then fills the gap with the correct nucleobases. After the DNA synthesis, the resulting nicked dsDNA is ligated [97,100,101].

DSBs are one of the most deleterious and toxic DNA lesions in cells. Eukaryotic cells have developed multiple pathways to repair these lesions in an error-prone and error-free manner [97,102]. NHEJ and HR are the dominant pathways for DSB repair. In human cells, NHEJ, which is error-prone, is the dominant pathway in the G1 phase, but RPA does not play a major role in NHEJ (for more details, see the review in [97]). In contrast, HR, a conserved and error-free process to repair DSBs, is the preferential DSB repair pathway in the S and G2 phases of human cells [102], and RPA plays a major role in this pathway (see below and [97]). To initiate HR at DSBs, the MRN (MRE11–RAD50–NBS1) complex recognises and binds the DSB followed by the recruitment of ATM and TIP60 (Tat-interactive protein, 60 kD, the catalytic subunit of the histone acyltransferase) to DNA [97,102]. The activated ATM phosphorylates H2AX, which in turn recruits MDC1 (mediator of DNA damage checkpoint protein 1) that is then phosphorylated by ATM. Next, the phosphorylated MDC1 recruits the ubiquitin E3 ligases RNF8 and RNF168, which together ubiquitinate H2AX [102]. Ubiquitinated chromatin sequesters 53BP1 (TP53-binding protein 1) and BRCA1 (breast cancer type 1 susceptibility protein). In the S/G2 phase, BRCA1 successfully counteracts 53BP1, initiating the ubiquitination of the downstream components such as CtIP and RPA (Figure 4 [97,102]). The ubiquitin ligase, RING finger and WD repeat domain 3 (RFWD3), ubiquitinates RPA70 and RPA32 at sites of DNA damage, which promotes HR [103]. In the following, RPA binds and stimulates the activity of BLM helicase and nucleases, e.g., DNA2 and EXO1, to unwind the damaged dsDNA and resect the 5′-ends of the DNA strands at a DSB site to create a 3′-overhang bound by RPA [97,103,104,105]. Phosphorylation of the N-termini of RPA70 and RPA32 inhibits the DNA resection in vitro and in cells [105]. The RPA–ssDNA filaments are finally transferred into RAD51–ssDNA filaments for the homology-dependent binding of a complementary DNA followed by DNA synthesis and resolution of the resulting DNA structure [97,103,104,105].

In the beginning of the DSB repair, MRN with its endonuclease activity and the help of CtIP starts the initial resection, whereas long-range resection occurs via BLM helicase/DNA2 or EXO1. In one resection pathway, BLM helicase unwinds dsDNA at DSBs in an ATP-dependent manner, and DNA2 resects the DNA in an RPA-controlled 5′–3′ directionality, as shown in Figure 4. MRN has a stimulatory effect by recruiting the BLM helicase to DNA ends and initiating the resection process [97,103]. The resection is terminated by the phosphorylation of RPA at the N-terminus of RPA32 and interrupting RPA–BLM interaction, causing a reduction in BLM helicase activity [10,103,105]. Alternatively, in the EXO1-dependent HR pathway, MRN recruits EXO1 to DNA ends and stimulates its 5′–3′ digestion process. This MRN stimulation is increased in the presence of RPA. The addition of the BLM helicase further stimulates EXO1 activity. This EXO-catalysed resection is also used in the mismatch repair of DNA [103]. Additionally, to stimulate and direct the resection reaction, RPA coats the 3′ ssDNA overhang, producing an RPA–ssDNA filament. Next, these RPA-bound ssDNA overhangs are exchanged to RAD51–ssDNA filaments to form the active RAD51 recombinase and to search for the complementary DNA [104,107,108,109]. RPA’s higher ssDNA affinity creates a barrier for the exchange from RPA to RAD51 on the ssDNA. It is important to note that RPA has ssDNA affinities of K_D_ = 1–20 nM depending on the ssDNA sequence composition (see above), whereas RAD51⋅ATP binds to ssDNA with an affinity KD = ~200 nM [1,6,8,9,19,110]. So, the replacement of RPA by RAD51 needs mediator factors such as BRCA2 in mammals and RAD52 in yeast [18,104,106,107,109]. 

In mammals, BRCA2 (breast cancer type 2 susceptibility protein) mediates the transfer of RPA–ssDNA intermediates to RAD51–ssDNA sequences where BRCA2 supports the nucleation of RAD51–ssDNA filaments, yielding a protein–DNA structure able to search for homologous DNA [18,104,106,107,109]. Although BRCA2 does not directly physically interact with RPA, its partner DSS1, which contains a solvent-exposed acidic domain, binds to RPA using this acidic domain to mimic DNA [106]. DSS1 performs multiple physical interactions with the DBD-A, B, C and F and thus interferes with the ssDNA-binding capability of RPA70. This reduces the affinity of the RPA to ssDNA and allows BRCA2 to load RAD51 onto the 3’-overhang ssDNA, replacing RPA [106]. Recently, Bell et al. determined that the BRC repeats of BRCA2 pre-assembled RAD51 ‘nuclei’, accelerating RAD51 nucleation [109]. Post-translational modifications have multiple functions in the transfer from RPA–ssDNA to RAD51–ssDNA filaments. It has been suggested that an extensive poly(ADP-ribosyl)ation of RPA reduces its high ssDNA affinity and that RAD51 preferentially binds to phosphorylated RPA [2]. Interestingly, the poly(ADP-ribosyl)ation of RPA enhances the recruitment of repair proteins to DNA damage sites [11,92]. Additionally, post-DNA damage, the SUMOylation of RPA70 causes a favourable binding of RAD51 to RPA [2]. In the presence of incorrectly functioning BRCA2, RAD51 is unable to replace RPA on ssDNA. The accumulation of RPA on ssDNA will result in a continuous activation of ATR signalling and downstream phosphorylation, including the N-terminus of RPA32 [2,97,105]. The phosphorylation of RPA32 in turn will inhibit the DNA resection at DSB carried out by a BLM helicase together with DNA2 and EXO1 [105]. The findings suggest feedback mechanisms and an interplay between DNA resection processes and RPA phosphorylation [26,105]. Correct HR completion is essential, since the dysregulation of HR by WASp deficiency can cause genome instability [96]. Importantly, the availability of functional RPA in the cells ensures that DNA undergoes HR instead of the error-prone microhomology-mediated end joining (MMEJ) where translocation and deletions may occur [17,111,112].

In yeast, RAD52 mediates the exchange of RPA–ssDNA to RAD51–ssDNA filaments. When RPA binds to ssDNA, the individual DBDs are in a variety of dynamic conformational ssDNA-bound states [2,14,18]. In the RPA–ssDNA complex, this flexibility allows access to the 5′- and the 3′-segment of the RPA–ssDNA complex. During the HR process, the recombination mediator RAD52 interacts with RPA bound to the 3′-overhang ssDNA of a DSB via the RPA32 WH domain [18]. This RAD52–RPA32 interaction specifically changes the ssDNA-binding dynamics of DBD-D and reduces the affinity of RPA to ssDNA, which in turn allows the nucleation of RAD52-bound RAD51 on the RPA-bound ssDNA. This further decreases the binding capacity of RPA to ssDNA and finally results in a RAD52-mediated removal of RPA and the exchange of RPA–ssDNA filaments into RAD51–ssDNA filaments, which are the HR-active components. These RAD51 filaments now will start to search for homologous DNA sequences on dsDNA located in close vicinity [18]. The RAD51 recombinase with the associated ssDNA having a free 3′-OH invades a nearby dsDNA and forms a D-loop [97,113]. In the following, RAD51 is removed, and the 3′-OH end bound to a template DNA serves as a primer for synthesis by Pol δ, κ and ν. The resulting DNA structure is resolved in a well-defined process as reviewed by Chatterjee and Walker [97].

## 7. RPA and Health Prospects

Numerous neurodegenerative diseases are caused by nucleotide triplet repeat sequences, e.g., CAG repeats in the genomic DNA of humans [114]. The progression of diseases such as Huntington’s disease occurs from an incorrect or lack of excision of these triplet expansion mutations [42]. RPA supports processes that suppress triplet repeat expansion in Huntington’s disease mouse brains protecting cells from CAG-expanded repeats and in turn from certain neurodegenerative diseases. In contrast, the expression of Alt-RPA inhibits the correct repair of triplet repeat expansions [42]. Interestingly, both types of RPA are overexpressed in Huntington’s disease and spinocerebellar ataxia type 1 (SCA1) patient brains. However, Alt-RPA is strongly overexpressed in these brains, whereas canonical RPA is mildly overexpressed. In this context, canonical RPA prevents somatic repeats expansions, whereas Alt-RPA promotes them [42]. The latter worsens the disease outcome. Interestingly, SCA1 mice showed a reduced disease phenotype after RPA overexpression, supporting the link between RPA modulating neurodegenerative disease pathogenesis [42].

To establish new drugs against cancers that have become resistant to existing therapies including chemotherapy, Van der Vere-Carozza et al. have created RPA inhibitors that target its OB-folds [115]. RPA is a central player in eukaryotic DDR pathways, and high levels of RPA expression act as a negative biomarker for patient survival in smoking-related lung cancer [115]. Additionally, increased levels of RPA in cancer chemotherapy may serve as an adaptive process to protect cells against genotoxic stress, suggesting that having active RPA and a fully functional DDR, cells have a chance to survive DNA damage, e.g., in cancer chemotherapy, whereas when inhibiting both RPA functions and DDR, cancer-treated cells become sentenced to apoptosis or other death pathways. Thus, using a combination of RPA and DDR inhibition ensures that a high degree of cancer cells with damaged DNA resulting from established DNA-damaging chemotherapies is eliminated [115].

Telomeres at the end of chromosomes shorten with each cell cycle due to the primer removal on the lagging stand during chromosomal DNA replication [20,116]. To overcome this limitation, cancer cells and stem cells recruit the enzyme telomerase to telomeres for their elongation and supporting cell immortality [20,116]. As telomeres naturally contain an ssDNA sequence, cells need to prevent RPA binding to telomeres and DDR induction to maintain normal cell growth and to shelter cells from negative reactions [117]. The protein protection of telomeres 1 (POT1) is a part of the SHELTERIN complex and is involved in the telomere length control of cells [117]. In POT1 mutant cells, the RPA-dependent activation of ATR DDR causes a telomerase-mediated hyper-elongation of telomeres, supporting the immortalisation of cancer cells [117].

## 8. Conclusions

The central roles of RPA in a variety of metabolic processes show the importance of future research using different angles regarding RPA functions and cooperation with replication, repair and recombination proteins. The main f prospects will be the examination of RPA levels and/or post-translational modification as biomarkers and testing its inhibition to combat cancers and other diseases. Additionally, the structural biological insights into the various metabolic processes including RPA at replication forks and RPA in complex with DNA recombination mediators on ssDNA, to name a few, will significantly enhance our understanding of the roles of RPA and such can be harnessed for drug development. enhance the field and give insights into the mechanisms of these central reactions.

## Figures and Tables

**Figure 2 ijms-25-00588-f002:**
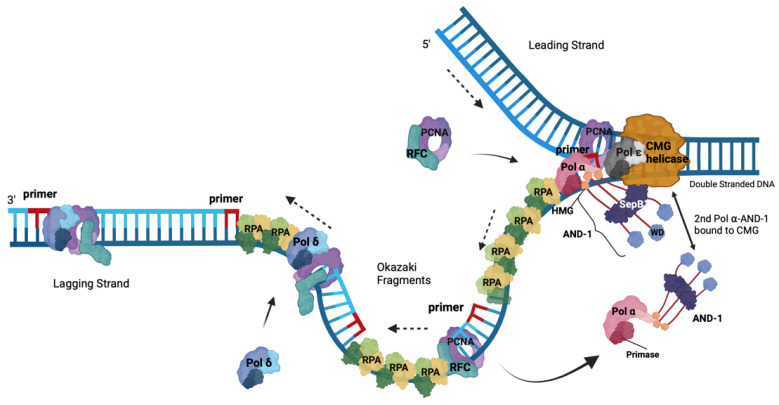
DNA synthesis at a eukaryotic replication fork. This schematic drawing of a eukaryotic replication fork shows CMG helicase (orange) unwinding dsDNA into the leading plus lagging strand templates and the replication proteins involved in DNA synthesis at the fork. Heterotrimeric RPA binds the resulting ssDNAs. The navy blue DNA strands are the parental strands. Pol ε (grey) newly synthesises the leading strand (royal blue DNA) in the 5′ to 3′ direction as indicated by the dashed arrow, and it is associated with the CMG. In contrast, the homotrimeric AND-1-CTF4-WDHD1 protein complex (dark grey, named AND-1) links CMG to the Pol α complex. The primer (red) synthesised by the primase function of Pol α allows for replication synthesis initiation by the DNA polymerase activity of Pol ⍺ (pink) in a 5′–3′ direction to start Okazaki fragment synthesis for lagging strand synthesis. PCNA (purple) and RFC (turquoise) replace Pol ⍺, making a landing site for Pol δ (maroon) to bind to the RNA–DNA and PCNA and elongate it until this complex reaches the next Okazaki on the parent strand, allowing the maturation of the Okazaki fragments to occur (adapted from [49,52,60,61]). The solid arrows represent incoming and leaving proteins.

**Figure 3 ijms-25-00588-f003:**
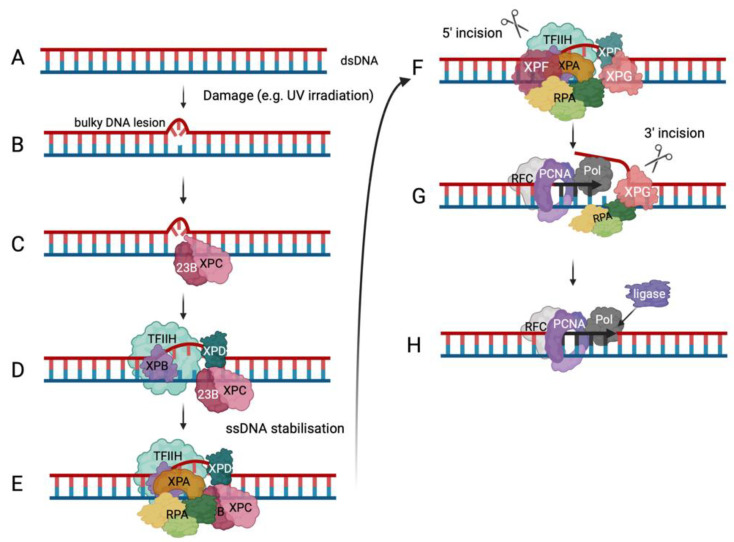
Nucleotide excision repair. The schematic drawing depicts the global genomic NER (ggNER) pathway as the most prominent NER pathway in cells [97,98,99]. Eukaryotic cells also repair bulky lesions via the transcription-coupled NER (tcNER) pathway, which only occurs in a transcription-dependent manner on the transcribed strand [97,98,99]. The tcNER pathway feeds into the presented pathway but was omitted for simplification reasons. (**A**) DNA damage can result in bulky DNA lesions. (**B**) RAD23B and XPC (pink) recognise these damaged sites in dsDNA. (**C**) They recruit the multi-subunit protein complex TFIIH (transcription facto II H, light green) that contains XPB in purple and XPD in dark green to verify and extend the opening of the dsDNA. (**D**) XPA, in orange, and the multi-coloured RPA are recruited to the DNA damage site to bind and protect newly unwound ssDNA. The positioning of RPA and XPA ensures the correct localisation of XPF and XPG, in darker pink colours, to undertake their incision functions as shown in the panel. (**E**–**G**) DNA elongation occurs after binding of RFC (grey) and loading of PCNA (purple) and DNA polymerase (dark grey) recruitment (panel **G**). (**H**) Ligase (purple) binds the newly synthesised DNA at the incision site and ligates the nick in the dsDNA to yield repaired dsDNA (adapted from Schärer, 2013 [99]).

**Figure 4 ijms-25-00588-f004:**
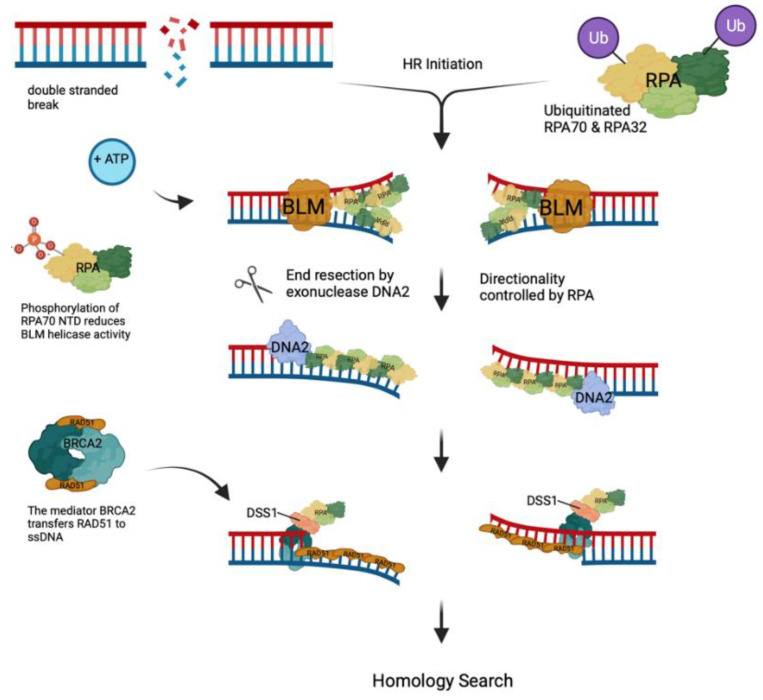
Homologous recombination. Double-strand breaks (DSBs) are highly toxic for cells and occur in DNA after high-energy radiation or cellular processes such as antibody development. Following DSB, the binding of MRN and chromatin modifications including the ubiquitination of cellular proteins such as RPA70 and RPA32 allows HR to occur (for more details, see the text). Then, BLM helicase (orange) unwinds the dsDNA in an ATP-dependent fashion and RPA binds to ssDNA. DNA2 (purple) interacts with RPA and degrades one DNA strand in the 5′–3′ direction controlled by RPA. The phosphorylation of RPA70 NTD alters the RPA–BLM interactions, causing a reduction in BLM activity [105]. The mediator BRCA2 via its partner DSS1 interacts with the resulting RPA–ssDNA filaments and exchanges RPA toward RAD51 [106]. The RAD51–ssDNA filament then starts a homology search of these overhangs (adapted from [103]).

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
