# Peer review of "Replication Protein A, the Main Eukaryotic Single-Stranded DNA Binding Protein, a Focal Point in Cellular DNA Metabolism"

_ijms, 2024, doi:10.3390/ijms25010588_

Round 1
Reviewer 1 Report
Comments and Suggestions for Authors
The review on RPA by Nasheuer et al. Reads well and is highly informative. I have mostly minor comments. I do think the structural aspects of RPA could be better presented. I have several specific suggestions below, but in general there is one structure shown (4GOP) from Ustilago maydis, but I imagine there must be several other structures that have been determined, and the authors could elaborate a bit more on what the different structural states are that have been elucidated, especially if there are structures in complex with any of the multiple partner proteins. Perhaps a table summarizing the different structures (with PDB codes, brief explanation) could be helpful, though that should be up to the authors – I realize this is not a structural review. In addition, the sections on replication and repair provide quite a bit of general information that is not related to RPA, and could potentially focus more on the details of RPA involvement.
Minor points
Line 13 “prevents the DNA sequence from nuclease attacks” perhaps replace “prevents” with “protects”
Line 43 Why say “apparent” molecular weights? Perhaps clarify that they are named after their apparent MW by SDS-PAGE (assuming this is the case). I take it that they were named prior to being sequenced (at which point the actual MW could be calculated).
Line 45. Instead of “coiled” beta-sheets perhaps “twisted” would be better.
Fig 1. The authors could color code the subunits somehow in panel A & B so that the two panels are coordinated better. The different subunits (or even domains) could be labeled in panels B and C. The colors of equivalent domains change again in panels D and E. The quality of the structural figures could be improved. The 5’ and 3’ ends of the ssDNA could be labeled.
Line 60, Fig. 1 legend, “zinc domains” could be “zinc-binding motifs”?
Lione 66-67, Legend for 1B, “Human STN1 and RPA32 OB-fold structures were predicted by alpha fold and are shown in light magenta and yellow, respectively”. I am confused by this sentence. What is STN1? (I don’t think it has been introduced yet). This is part of the 1B legend but Fig1b does not have anything yellow or magenta. Are the structures shown in panel B (cyan) not part of the experimentally determined structure? I am unclear why they are being overlayed in panel C.
Fig1D,E. All of the different domains should be labeled. This is apparently the human RPA alpha fold model, where as panels B,C are the experimentally determined structure of RPA from Ustilago maydis (this could be stated more explicitly in the legend). It is stated that the orientation of D,E is similar to B,C, but it is difficult to discern this from the figure. Labeling of the different domains would really help here.
Line 73, “DNA-F domain”, I think the authors mean “DBD-F” domain.
Line 878, the authors mention three different DNA binding modes. Which mode is shown in the structure of Fig1B,C? Do these different binding modes have functional relevance?
Line 98 – what length of polypyrimidine?
Fig 1B, Fig 1D – could the authors keep the color coding the same for the different domains? RPA32 and RPA14 are colored cyan and brown in panel B, but purple and orange in panel D. Please label the different domains on the actual figure (not just in figure legend). It is much easier on the reader that way.
Lines 115-119 – long run on sentence that doesn’t make sense.
DNA replication section – too much information about replication with little specific info on RPA. Focus should be on RPA.
Roles of FEN1, DNA2 in 5’-flap cleavage could be shown in part of Fig. 2.
Lines 378-379, “feeds into presented pathway at diagram”. Something wrong here not sure what was intended.
Line 458, what is “N.B. RPA”?
Line 545, “will significantly enhance field” could be “the field”
Comments on the Quality of English Language
Minor changes needs, some of which I have indicated (but not all)
Author Response
First, I want to thank the reviewer for the useful comments. The requests and suggestions were very helpful and included for in the revised version as discussed below. First, I want to apologise for the formatting errors in the headings. They were in the word document in a different format and then automatically reformatted yielding this strange mixture of capital and small letters. Unfortunately, I did not carry out a sufficient final proof-reading of the submission but only a general overview.
Regarding the experimental structures currently available, the included experimental structure published by Fan & Pavletich is the best and most complete experimental RPA structure (4GOP) so far. There are multiple substructures and domains also published but none is as complete as this structure. Recently, RPA filaments with RPA from a archaeal bacterium with low resolutions and missing several domains have been published but they were not helpful for further discussions and were not included. The structures were included to help a better understanding of RPA functions. The AlphaFocus-predicted structure that helps to visualise RPA domain organisation, which cannot be found in any experimental structure and is therefore helpful despite the limitations, e.g., the predicted structure shows how flexible some regions of RPA can be.
The review has been focus followed the reviewers suggestions e.g., the section 'Understanding the Initiation of Okazaki Fragment Synthesis at the Molecular Level' was omitted following the reviewers suggestion. However, to include a wider view of RPA's function additional information about DNA replication and repair is included. I hope that the reviewer can agree with the revised version submitted.
Minor points:
- the abstract line 15 (now line 13) the word 'protect' was used as highlighted. The last sentences of the abstract were also reworded.
- 'Line 43' (now line 46): the wording 'as determined by SDS gel electrophoresis' was added.
- 'Line 45' (now line 49): The word 'twisted' was used.
- The figure 1 was changed as suggested and the legend was also modified to include the changes.
- 'Line 60' (now line 102): The term 'zinc-binding motifs' was used.
- 'Line 66-67' (now line 112-113): STN1 was previously short mentioned (current line 88) but an additional comment is included regarding STN1 for clarity reasons. More details about STN1 in section 5.
- 'Fig1D,E.' The advice of the reviewer was taken on board and the figures were changed. The experimental and predicted structures were more clearly labelled in the figure legend.
- Line 73 (now line ) 'DNA-F domain': the figure legend was much reworded and the wording was corrected.
- Line 87(?) (now line 65): regarding the RPA-ssDNA complexes and different binding modes. - A sentence to explain some putative functional consequences following these flexibility in ssDNA binding is included (line 66-69). Further discussion then in section 6.2.
- Line 98 (now line 79): The length of the Poly(dT) was not described but it was a mixture of different length. The publication is very old but everybody uses these data for comparison. Therefore, some newer data were included. The paragraph has been reworded and is hopefully now acceptable.
- Fig 1B, Fig 1D colour-coding: The colour-coding of Figure 1 from panel A to panel E has been adjusted.
- Lines 115-119 (now line 134-142). The sentence has been reworded and split into several sentences.
- DNA replication section. The replication section was restructured and the section 'Understanding the Initiation of Okazaki Fragment Synthesis at the Molecular Level' was removed to streamline the replication section.
- 'Roles of FEN1, DNA2 in 5’-flap cleavage could be shown in part of Fig. 2.' The suggestion to include the maturation step in more detail e.g., showing the roles of FEN1 and DNA2 would make the figure more complete but would also result in very complicated and complex figure, which would be difficult to follow for a general reader. Therefore, the authors suggest to keep the figure as it was.
- Lines 378-379 (now 408-409). The sentence was reworded.
- Line 458 (now line 495). The phrase 'N.B.' was reworded to 'It is important to note ...'
- Line 545 (now line 589). The sentence was changed as suggested.
Reviewer 2 Report
Comments and Suggestions for Authors
The article titled “Replication protein A, the major eukaryotic single-stranded DNA-binding protein, a focal point in cellular DNA metabolism” by Nasheuer et al. is a review that takes stock of the situation regarding the human single-strand binding protein and its importance in DNA metabolism.
The topic is notable, the review covers all the knowledge about it well, but there would be many things to correct.
However, I noticed several errors in terms of typing and proofreading accuracy, but also small scientific inaccuracies
Below are some examples
Typing and proofreading errors
Many paragraph titles contain case errors (capital letters/lowercase):
3. ROLES OF RPA IN DNA Replication;
4. understanding the INitiation of okazaki fragment synthesis at the molecular level
5. RPA and DNA rePAIR PROCESSES
6. RPA IN DNA Damage signalling
7. RPA IN DNA repair pathways
In figure 4 on line 64 it is written “panel B”, while on line 71 it is written “Panel D and E”.
Probable copy-paste errors:
Line 434 “develop-ment”
Line 436 “un-winds”
Line 479 “de-phosphorylation”
Lane 459 “RAD51•ATP” (very big dot)
Line 366: before “In the global genome…” there are too many spaces.
Line 470 "et al" instead of "et al." (latin word)
Line 518 "et al" instead of "et al." (latin word)
Lane 182: “de novo” instead of "de novo" (latin word)
I also strongly recommend writing all the acronyms (in the text or in the "Abbreviations" paragraph) to make reading easier for everyone, even those who are new to the topic. See for example lane 421 (TIP60), lane 422 (MDC1), lane 424 (BRCA1)…
Small scientific inaccuracies
Lane 182: “No DNA polymerase starts DNA synthesis de novo”. The paper entitled “De novo DNA synthesis by human DNA polymerase lambda, DNA polymerase mu and terminal deoxyribonucleotidyl transferase” (Ramadan K, et al. J Mol Biol. 2004 May 28;339(2):395-404. doi: 10.1016/j.jmb.2004.03.056. PMID: 15136041) suggests that this activity is possible, at least in vitro. I suggest to modify this sentence.
Lane 154: "MCM2-6" maybe you meant "MCM2-7"?
Overall, I believe the manuscript will be of interest to IJMS readers and could be published in this journal. Therefore, after careful revision by the authors themselves, I believe it could be considered by the Journal

Author Response
The authors want to thank the reviewer for their thoughtful feedback. Unfortunately, during the submission process, the original manuscript was significantly reformatted and the headings as well as the numbering was changed. The authors apologise that these changes were then not picked up in the proof-reading process. In the revised manuscript, the formatting errors have been corrected.
After the submission of the manuscript, the role of Alt-RPA was discussed with a colleague in the triplet repeat field. Since Alt-RPA has been neglected in the RPA field and beyond but new findings will raise interest the paragraph was significantly re-written and included as a special section in the revised version. I hope the reviewer can agree with these changes.
All changes have been highlighted in yellow in one version of the resubmission.
Corrections:
- 'Many paragraph titles contain case errors (capital letters/lowercase)': these formatting errors have been corrected.
- 'In figure 4 on line 64 it is written “panel B”, while on line 71 it is written “Panel D and E”.' Feedback: The legend of figure 1 was significantly re-written and the errors corrected.
- 'Probable copy-paste errors': these errors have been corrected.
- 'Line 366: before “In the global genome…” there are too many spaces.' The error has been corrected.
- The formatting errors in Line 470, Line 518, and Line 182 (now line 507, 560, and Line 164) were corrected.
- All the acronyms (in the text or in the "Abbreviations" paragraph) have been annotated ion the text and in the Abbreviation table.
- Line 182 (now line 226): The sentence was reworded to 'No replicative eukaryotic DNA polymerase starts ...'
- In the revised text 'MCM2-6' was changed to 'MCM2-7'.
Reviewer 3 Report
Comments and Suggestions for Authors
SUMMARY:
Nasheuer et al. examined the roles of replication protein A (RPA) in cellular DNA metabolism. The review began with an exploration of RPA's characterization, its interaction with single-stranded DNA (ssDNA), and its crucial associations with other proteins. The subsequent sections delved into RPA's functions in DNA replication, the DNA damage repair process, and DNA damage signaling.
Primarily, the authors undertook significant efforts to consolidate and summarize the intricate mechanisms of DNA replication and DNA damage repair pathways. And they systematically discussed the important roles of RPA involved in these processes. Nevertheless, there is a need for a restructuring of the paper to enhance overall clarity before publication.
Comments:
1. The outline concerning RPA in DNA repair appears confusing. Sections "7. RPA in DNA repair pathways" and "8. RPA and DNA recombination" should ideally serve as subsections within the framework of "5. RPA and DNA repair processes." Please consider reorganizing the content to better delineate the roles of RPA in DNA repair.
2. I am a bit confused about the section “4. Understanding the initiation of Okazaki fragment synthesis at the molecular level”. Some of the context is related to transcription described in section 3, while other parts are about telomere metabolism. To enhance clarity, could the authors consider creating an independent section to describe RPA's role in telomere metabolism?
3. The authors outlined the primary roles of RPA in DNA homologous recombination. However, some recent discoveries concerning RPA in DNA resection have not been incorporated. For instance, a recent paper shows that the interaction between RPA and BLM enhances processive resection by EXO1 and DNA2 nuclease, and RPA32 phosphorylation inhibits DNA resection (PMID: 31153714).
4. Please correct the “RPA2” in line 480.
5. There are some typos present in the context. For example, “INitiation” in line 241, “rePAIR” in line 308, “pre-sented” in line 378, recombina-tion” in line 433, “develop-ment” in line 434, “un-winds” in line 436 etc. Please correct these, along with any other typos that may be identified.
Author Response
Firstly, the authors want to thank the reviewer for their thoughtful feedback. Unfortunately, during the submission process, the original manuscript was significantly reformatted and the headings as well as the numbering was changed. The authors apologise that these changes were then not picked up in the proof-reading process. In the revised manuscript, the formatting errors have been corrected.
After the submission of the manuscript, the role of Alt-RPA was discussed with a colleague in the triplet repeat field. Since Alt-RPA has been neglected in the RPA field and beyond but new findings will raise interest (Alt-RPA and its involvement in the development of neurodegenerative diseases) the paragraph was significantly re-written and included as a special section in the revised version. I hope the reviewer can agree with these changes.
All changes have been highlighted in yellow in one version of the resubmission.
Response to the comments of the reviewer:
- The review was restructured and hopefully has now a clearer view. Following the comments of reviewer 1 the replication section was shortened and the first part of the section 'Understanding the initiation of Okazaki fragment synthesis at the molecular level' was omitted.
- The remaining section with the heading 'The Role of the RPA-Related, ssDNA-Binding Protein Complex CST in the Initiation of Okazaki Fragment Synthesis' focuses on the similarities of STN1 and RPA32 in DNA replication. The authors hope that the revised structure of the review is clearer and the contents is easier to understand.
- The recombination section was significantly revised following the reviewer's suggestions and including the recommended literature.
- RPA2 (line 480) is corrected and changed to RPA32 (line 520).
- The typos were identified and corrected.
Round 2
Reviewer 2 Report
Comments and Suggestions for Authors
The article titled “Replication protein A, the major eukaryotic single-stranded DNA-binding protein, a focal point in cellular DNA metabolism” by Nasheuer et al. is a review that takes stock of the situation regarding the human single-strand binding protein and its importance in DNA metabolism.
The topic is notable, the review covers all the knowledge about it well.
Formatting errors have been corrected
Moreover, the authors accepted all suggestions according to other reviewers. For these reasons, I think that the paper is suitable for publication in the International Journal of Molecular Sciences
Author Response
Thanks for your kind comments.
Best wishes,
Heinz Nasheuer
Reviewer 3 Report
Comments and Suggestions for Authors
Response to Nasheuer et al. regarding their reply on the manuscript ijms-2776184:
The authors have reorganized the manuscript by shortening the replication section, introducing an individual Alt-PRA section, and restructuring the outline of RPA in DNA repair and DNA damage signaling. Adding an individual section for Alt-PRA is a good improvement, given its growing significance in the field. The recommended literature has been incorporated into the recombination section. The revised manuscript is structurally clearer and easier to follow. I just have minor comments before accepting the publication of the manuscript.
Minor comments:
1. Identified a few typos, such as the missing "bracket" in "SDS gel electrophoresis (Figure 1, [1, 5, 12]." in line 47, and "N-termini of PRA70 of RPA32" in line 463.
2. In lines 523-526, the authors stated that " Importantly the availability of RPA in the cells ensures that DNA undergoes HR instead of the error-prone microhomology-mediated end joining (MMEJ) where translocation and deletions may occur [2]. " To my knowledge, the availability of RPA itself does not determine the repair pathway choice. According to ref. [2], they described that “the RPA-ssDNA platform is extensively phosphorylated, SUMOylated and ubiquitinated in response to damage. These post-translational alterations of RPA regulate the activation of the ATR checkpoint and also promote DNA repair, particularly through the HR pathway. ” and “post-translational modifications of the RPA complex may direct the specific functions of RPA-ssDNA in different sub-pathways of the DDR.” I would like to advise the authors to be more careful and rigorous for their descriptions.
Author Response
The correction according to reviewer were included. Unfortunately, there was a mix up with the references and the correct references [17, 111, 112] instead of reference #2 were included.
Many thanks for the feedback and the help.
Best wishes,
Heinz Nasheuer